# The Human Family—Its Evolutionary Context and Diversity

## Karen L. Kramer

Department of Anthropology, University of Utah, Salt Lake City, UT 84112, USA; karen.kramer@anthro.utah.edu

**Abstract:** The family defines many aspects of our daily lives, and expresses a wide array of forms across individuals, cultures, ecologies and time. While the nuclear family is the norm today in developed economies, it is the exception in most other historic and cultural contexts. Yet, many aspects of how humans form the economic and reproductive groups that we recognize as families are distinct to our species. This review pursues three goals: to overview the evolutionary context in which the human family developed, to expand the conventional view of the nuclear family as the 'traditional family', and to provide an alternative to patrifocal explanations for family formation. To do so, first those traits that distinguish the human family are reviewed with an emphasis on the key contributions that behavioral ecology has made toward understanding dynamics within and between families, including life history, kin selection, reciprocity and conflict theoretical frameworks. An overview is then given of several seminal debates about how the family took shape, with an eye toward a more nuanced view of male parental care as the basis for family formation, and what cooperative breeding has to offer as an alternative perspective.

**Keywords:** behavioral ecology; family studies; cooperative breeding; patrilineal





## 1. Introduction

Family formation shapes many aspects of our daily lives—who we work, eat and sleep with, who we share with, whether we live in large extended or conjugal families, and the time males and females, adults and children, spend in the company of each other. The ways families are structured also affect where we live after marriage, who we inherit our name and property from, who is included in our kin group and who is excluded, whether we marry one or multiple partners, how disagreements and authority are brokered, whether both females and males can instigate divorce and the extent to which gender equality prevails. The family expresses a wide array of forms across cultures and ecologies, and within any one society. While the nuclear family is the norm today in developed economies, it is the exception historically and in many other cultural contexts. Indeed, if there is any way to characterize the human family, it would be its diversity and flexibility.

This article reviews several avenues of research in behavioral ecology that have addressed the evolution of the family. First, I provide an orientation to characteristics that distinguish group and family formation in the human lineage from other closely related species. This is followed by an overview of the theoretical contributions that behavioral ecology has made toward understanding dynamics within and between families; these include life history, kin selection, reciprocity, the division of labor and other cooperation and conflict frameworks. Some of the important debates that have emerged from research on the evolution of the family are then discussed. This review has three aims: to discuss what is known and is speculative about the ancestral context in which the human family evolved; to recast conventional views of the nuclear family to reflect the empirical, cross-cultural record; and offer alternative perspectives to the patrifocal tradition of describing the human family.

Throughout, I draw on examples from contemporary, small-scale societies (also called traditional societies) for several reasons. In behavioral ecology, the topic of this Special Issue on the family, small-scale societies, particularly hunter-gatherers, have been central to study

because their demographic and subsistence conditions, and hence social lives, encompass more diverse forms of the family than are often evident in industrialized societies.[1] Family life has substantially changed in recent centuries with urbanization and industrialization and is novel in many regards. In industrialized societies, conjugal families (spouse(s) and their dependent children; also called the nuclear family) are the norm. The reduction in fertility in most of the developed world means that children live in small families with few siblings. Families are not only smaller because of the multigenerational effects of the demographic transition and longer generational times, they are also composed of fewer collateral kin (aunts, uncles and cousins). Because of high rates of divorce, remarriage, and geographic dispersion, nuclear families are often isolated from grandparents and other relatives. On an evolutionary time scale, this trend toward atomization into small conjugal groups is quite recent; for most of human history, society was seldom organized as such (Van den Berghe 1990). It is important to clarify that small-scale societies are not thought of as relics of the past, but exemplify a more representative, diversified and inclusive view of human family life. That said, many of the tenets presented here are directly applicable to family formation in industrialized societies, and understanding the novel constraints and opportunities that nuclear family organization has presented.

Where necessary, specialized language is italicized and briefly defined. As a review of behavioral ecology approaches to family formation, the charge here is to hopefully communicate the usefulness of this approach to family studies in the social sciences generally.

### 1.1. Characterizing Social Structure in the Human Lineage

Nonhuman primates offer a comparative lens to appreciate those social and family traits that are part of a common primate heritage, and those traits that are particular to our species and derived in the human lineage. Chimpanzees, the great ape genetically most closely related to humans, have long been used as the behavioral model assumed to best resemble the mating and childrearing structure of the deep past. More recently, however, this has given way to debate about whether our ancestors lived in multimale-multifemale polygynandrous (both sexes mating with multiple partners) groups such as chimpanzees (Gavrilets 2012; Hrdy 2009; Van Schaik and Burkart 2010), or were instead organized in polygynous, gorilla-like harems (Dixson 2009; Grueter et al. 2012), or had a hamadryas baboon-like structure with multiple single-male groups living together within a larger population. In fact, certain family and social characteristics may have more in common with some species of birds (Van den Berghe and Barash 1977), social carnivores and even social insects (Moffett 2019), than chimpanzees with whom we share much of our genetic makeup. Despite debate over the social organization from which the hominin line developed, most researchers agree that group living and multilevel societies are ancient features of human sociality.

Humans can be broadly described as living in multilevel societies organized in nested interacting levels (Grueter et al. 2012; Hamilton et al. 2007; Chapais 2008, 2011; Kelly 2013; Marlowe 2005a; Flinn et al. 2007), including conjugal families, extended families, multi-family residential clusters, bands, tribes, with layers added as political, demographic and hierarchical complexity increase. In principle, the number of nested levels is unlimited (Chapais 2013), including church, state, national and global institutions in contemporary industrialized societies. In contrast, small-scale societies are usually characterized as having local autonomy and authority, and truncated interactions with centralized or top-down institutions (see Note 1). Marriage, dispersal, provisioning and childrearing patterns, as discussed in the following sections, shape family formation, and are the source of its variation and relationship to more inclusive social entities. It is important to point out, however, that any generalized formulation of human social structure, or characterizations of family living in traditional and industrialized populations have numerous exceptions.

### 1.2. Human and Nonhuman Primate Multilevel Systems

Human societies are comprised of what can be described as multimale-multifemale groups with multiple breeding females. This combination is rare in animal societies—the explanation usually given that rivalries between males competing for females prohibit social cohesion. Although societies composed of multiple breeding males and females are found among a few other primates, including chimpanzees and some baboons, several features distinguish the structure of human communities.

First, humans reside in multifamily residential clusters formed around long-term pairbonds (Chapais 2013). In all human societies, pairbonds are socially recognized through marriage unions, which have a range of monogamous (one male/one female), polygynous (one male/multiple females) and polyandrous configurations (one female/multiple males) that vary both within and across societies (Apostolou 2007; Beckerman and Valentine 2002; Flinn and Low 1986; Marlowe 2000; Walker et al. 2011). Pairbonds are certainly not the only form that intimate relationships take, and in no society are sex and parenthood likely restricted to marriage. The point here is that pairbonds exist in all human societies. They may have been favored for a variety of reasons, but are generally thought to be derived in the hominine line (Quinlan 2008).

Second, across human societies, men and women, adults and children do different tasks, target different resources and share the fruits of their labor. While this takes many forms and details vary widely cross culturally, the age and sexual division of labor is foundational to human subsistence (Alvard and Nolin 2002; Codding et al. 2011; Gurven 2004a; Gurven and Hill 2009; Kaplan et al. 1990; Kuhn and Stiner 2006) and childrearing (Hrdy 2009; Kramer 2011), two pillars of family formation. While the age and sexual division of labor is not unique to humans, the combination of pursuing different subsistence activities, cooperating in joint activities, sharing childcare, food, and other resources is unmatched among other primates.

Third, adults maintain often life-long relationships with their natal families and move easily between residential groups. This fluidity is an unusual primate trait, and serves to establish social networks across residential groups (Chapais 2008; Grueter et al. 2012; Rodseth et al. 1991). This has been extensively studied in hunter-gatherers, who form bands ranging in size from 35–80 adults and children, comprised of families of various descriptions, within which members cooperate in daily subsistence and childrearing activities (Gurven 2004a, 2004b; Hamilton et al. 2007; Marlowe 2005a, 2005b). Bands form more inclusive social entities who share the same dialect, communal access to resources, and gather occasionally for purposes of ritual, politics, trade, exchange information, gifts, mates, sports or warfare (Kelly 2013). This is unlike anything other great apes do.

Among chimpanzees, for example, males are the philopatric (staying in one's natal group after sexual maturity) sex and are highly antagonistic, sometimes lethally, toward males from other troops (Boesch and Boesch-Achermann 2000; Goodall 1986; Nishida et al. 1985; Watts and Mitani 2001; Wrangham 1999). Chimpanzee females, usually the dispersing sex, likewise often encounter acrimony when they join a new troop, tend not to form close female bonds (Gilby and Wrangham 2008; but see Lehmann and Boesch 2009), and are unlikely to see their mother or siblings after they leave their natal troop at maturity. Explanations for why humans are so varied in their dispersal and residence patterns and cultivate social bonds across multiple groups have centered on building networks to exchange food, raw materials, labor (Gurven 2004b; Gurven and Hill 2009; Hamilton et al. 2007; Hill and Hurtado 2009; Hill et al. 2011; Kaplan and Hill 1985; Kaplan et al. 2000; Kramer and Greaves 2011; Kaplan et al. 2000), information (Binford 2001) and marriage partners (Kramer et al. 2017).

Ties across multiple groups are possible because humans recognize both maternal and paternal relatives and maintain relationships with their natal kin after dispersing. This interacting social structure where individuals can move between groups rests on the vast relationships that humans keep track of, a dexterity feasible only with spoken language. The complex kin terminologies, which are foundational in traditional societies,

allow people to monitor relations across both generations and over large geographic areas (Meggitt 1962; Berndt and Berndt 1964; Schefler 1978).

### 1.3. Characterizing the Human Family

What then of the family? Defining the family is elusive in a "you know it when you see it", but a difficult to delimit way. Common considerations from sociology, anthropology and psychology are the family as a lineage, or line of descent; a grouping of a consanguineal (blood relatives) and affinal kin (in-laws), the basal economic unit or unit of production and consumption, and the group of people that live together in the same structure or property. Many researchers place marriage and kinship at the organizational core of the family, with marriage assembling not only partners who cooperate, but networks of kin and their associated alliances, obligations and responsibilities.

Ethnographic studies give an insider's view of how people assort in small-scale societies. Even so, circumscribing the family is not without its challenges. Ethnographers can ask and readily get answers to questions such as who are your parents, grandparents, children and siblings, and from this information construct kinship and relatedness trees. One can also ask and observe who shares with whom, who works with whom, who cares for children, who occupies a residential structure or compound, and a range of more qualitative questions. But how to identify a family as a unit with discrete membership is less straightforward. For example, in the Pumé language (hunter-gatherers living on the llanos of Venezuela), there is no equivalent term for family (Box 1). When away from camp and a Pumé encounters another, he or she is greeted first with a general kinterm, roughly translated as 'my female relative' or 'my male relative', which signifies their identity as Pumé, and then specifically through kin ties, by association through parentage, siblings or children. The Pumé do not use individual names to address each other and kin terminology is sufficiently complex to reference the 500 or so band members that they might encounter over their lifetimes. But no moniker exists for 'my family'.

As a root grouping, the conjugal family is identifiable through biological parentage, and the ethnographer can record the composition of residential units and sharing groups (often called households). But membership becomes blurry at the edges because the nuclear family in most societies outside of the industrialized world is seated in a gradient of relationships—extended families or joint families, which are nested in larger entities that interact to different degrees, depending in part on geography and propinquity.

While definitions are seldom satisfactory, and it is difficult to observe the family in isolation from other social entities, how the family is circumscribed does become important in analytic and modeling decisions. People assort in a range of groupings depending on whether the aim is parenting, meeting subsistence needs, or joint large-scale collaborative projects. The units of analyses might then be expected to differ with the research question.

**Box 1.** Savanna Pumé marriage and family.

The Savanna Pumé are a group of mobile hunter-gatherers, who live on the llanos of west-central Venezuela (Kramer and Greaves 2017; Kramer et al. 2017), and are typical of other tropical foragers in many aspects of their social and family lives (Marlowe 2005a, 2005b). They live in small, multifamily groups of ~60 adults and children. While polygyny occurs, most marriages are monogamous (13% of men and 9% of women have been polygynously married at some point during their lives). Individual marriages vary, with some couples having life-long monogamous marriages (64% of marriages), while others may marry polygnously for some years, divorce and remarry several times. Both males and females have autonomy when and whom to marry, divorce and remarriage decisions. Couples are considered married if they are sexually engaged, and many adults (40%) reenter the marriage market after their first marriage dissolves due to either spousal death or divorce. Girls have their first children on average in their mid to late teens, and have 5 to 6 children over the course of their reproductive years. Child mortality is quite high, with many children not surviving the hardship of their first wet season (Kramer and Greaves 2007). Household composition ebbs and flows in annual cycles. Multiple families aggregate during the wet season into long-houses and disaggregate into separate brush shades during the dry season. Sharing relationships and work parties likewise are structured differently at different times of the year, and for different resources (Figure 1). An individual's relationship to the larger group of Savanna Pumé is accorded through kinship, which include terms for closer and more distant relatives, consanguineal and affinal distinctions, older and younger siblings, ascendant and descent kin. Many Pumé maintain a life-long affiliation with a particular band, and frequently visit relatives who live in other bands.

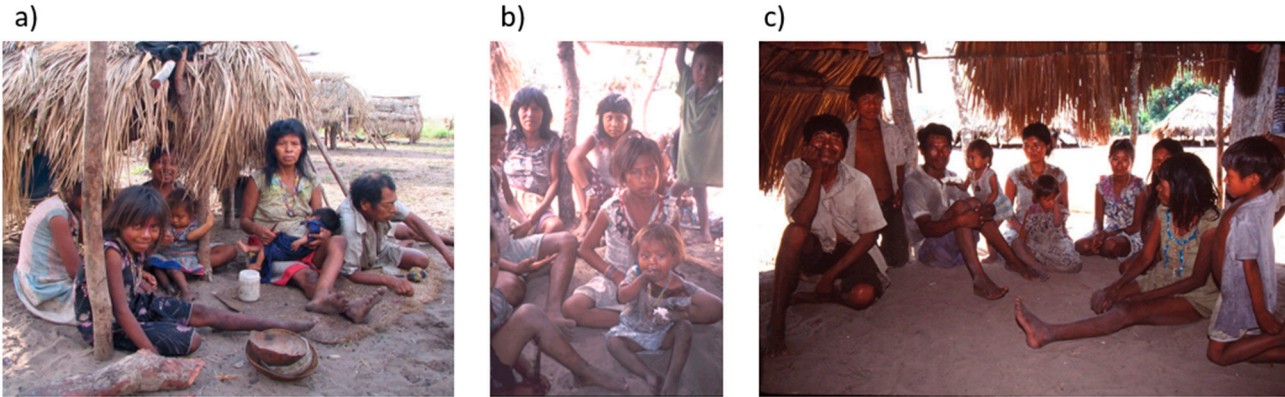

**Figure 1.** Three Savanna Pumé group views: (**a**) a conjugal family, (**b**) a female foraging party group composed of kin and nonkin from several families and (**c**) an extended family.

### 1.4. Ancestry of the Family

As a broad historic trend, the family has diminished in size, shape and function as many of its economic, educational and religious roles are replaced by church and state. The family as a conjugal unit is quite recent and prevalent only in some parts of the world (Sear 2016). For example, prior to the 18th century, European languages did not include a term for the nuclear or biological family. The Roman word *familia*, equivalent to house, referred to the residential group, but no specific term was given to the parent–child unit (Gies and Gies 1987). But what is known of the family in the distant past?

Most animals are solitary except for forays to mate or raise offspring until they fledge or wean. Getting together to mate or parent may be transient or enduring, depending on species. Although the family leaves no unambiguous fossil or archaeological record, living great ape social organization gives clues about the evolutionary hurdles that were overcome for multifamily groups to emerge and live together in relative amicability.

If the ancestral hominin society was chimpanzee-like with a multimale-multifemale group composition, the challenge would have been to establish stable mating bonds within a polygynandrous breeding system. For both males and females to 'agree' to pair-bond, ensure paternity and biparentally invest in offspring is no small feat, which is why as a family system it is so rare in mammals (Clutton-Brock 1991; Lukas and Clutton-Brock 2012). If the ancestral system was gorilla-like instead (Geary and Flinn 2001), the transition would have involved coalescing single- or multi-male harems into multifamily communities, which would require an equally tough shift in dampening male competition, constraining polygyny and developing mutual regard for each other's pairbonds. Either scenario likely occurred over millions of years and in multiple stages (Chapais 2013).

Less often considered is the possibility that family systems emerged from matrifocal groups. Elephants, whales, lions and some social carnivores are examples where mothers benefit from raising young in nursery or crèches groups (Lukas and Clutton-Brock 2012). In these communal breeding species, it is advantageous for nursing mothers to stick together because young are more protected from predators in larger groups, rather than to share care or feeding per se. In humans, cooperative interactions between mothers and offspring extend well beyond weaning, and the formation of groups of mothers and her juvenile children or of multigenerational mothers may be a critical but understudied step in the evolution of the human family (Kramer 2011, 2014; Kramer and Otárola-Castillo 2015).

## 2. Behavioral Ecology Approaches to the Family

Behavioral ecology developed within the field of evolutionary biology in recognition that behavior, as well as biology, is shaped by natural selection. Its application to humans provides a theoretical and empirical basis to evaluate those aspects of social and family structure common across societies due to a common evolutionary past. The emphasis on the interaction between ecology and behavior, and consequently on phenotypic plastic-

ity (West-Eberhard 2003) also gives a framework to generate predications about family variation across environment, history and culture.

### 2.1. Life History Theory

The diversity of marriage, family and kin arrangements can be seen as varied responses circumscribed by a shared biology and life history. Life history theory, an integral component of behavioral ecology, views species diversity as the outcome of different ways to allocate time, resources and energy across the life course to maximize fitness. Having a hybrid of both a slow and a fast life history shaped the central problem human mothers face, to which the family, at its simplest a small cooperative group, is a solution.

Because humans grow slowly, mature late and live long lives, they are often characterized as having a slow life history (Charnov and Berrigan 1993; Gurven and Walker 2006; Walker et al. 2006; Bogin 2006). As a species-specific pattern, human children take two decades before they start their own reproductive lives, which is long for a primate. In addition to growing slowly, human children are more likely to survive. In hunter-gatherer societies, a child is almost 40% more likely to survive to reproductive age than a chimpanzee juvenile (Gurven and Kaplan 2007; Hill et al. 2001; but see Wood et al. 2017). But with some interesting nuance. Whereas little difference is evident between hunter-gatherer and chimpanzee infant survivorship (both are ~80%), gains in juvenile survival (from weaning to reproductive age) differ markedly. Although variation exists among studies, a human forager is almost twice as likely to survive to age 15 than a chimpanzee. Most explanations for the higher human probability of survivorship point to the effects of caring for juveniles. Among other great apes, after weaning, food sharing is minimal between adults and offspring, and juveniles provision their own calories. Along the hominin line, a different life history strategy was favored; infants were weaned early and young juveniles were fed.

Once mature, natural fertility (where women do not use parity-specific birth control) mothers make up for late maturity and a slow start by reproducing quickly (what would be called a fast life history). Nonhuman great ape mothers nurse their young on average for 4–6 years (Robson et al. 2006; Thomson et al. 1970). Comparatively, in a society without bottle feeding, mothers fully wean babies between the ages of 2 and 3 (Kennedy 2005). The relatively young age at weaning is closely associated with short birth intervals. In natural fertility societies, children are spaced on average ~3.1 years apart, a birth interval two to three times shorter than that of other great apes (Kaplan et al. 2000; Lancaster et al. 2000; Thompson et al. 2007). Combined, a hybrid life history of a slow maturation and a *fast-reproductive* pace commit mothers to raise multiple dependents of different ages—something a nonhuman great ape mother rarely does (Lancaster 1997). This evolving life history, which took millions of years, at some point in the past would have posed a time allocation and economic problem for mothers (Kramer and Otárola-Castillo 2015); how to find the time and resources to care for multiple dependents of different ages? How this was solved in many ways is what sets the human family apart from other primates.

### 2.2. Kin Selection and Dynamics within Family

The primacy of kin relations to family formation has a long history of study in anthropology and the social sciences. Kin selection theory, developed in evolutionary biology in the 1960s (Hamilton 1964; Smith 1964), added a genetic logic to the centrality of kin relations in social structure. Kin systems can be seen to serve two main functions: identity and cooperation.

Family formation at its most fundamental rests on being able to recognize one's parents, siblings, and other relatives (Box 2). In addition to those closest to us, humans are amazingly astute in their capacity to identify kin and have boundless ways to codify familial membership. For example, griots of Malian West Africa preform genealogical recitations, which may reach back many generations, as a way to legitimatize relatedness (Irvine 1978). For most mobile hunter-gatherers, kin identity carries important information

when meeting strangers, allowing them to communicate their connection, in a sense as a letter of introduction, to gain access to resources or territory outside one's home range.

Kin identification, enabled through language and elaborate kin terminology, concretizes in and out groups to establish rules for marriage and incest avoidance. In this regard, the diversity of marriage and residence norms is neither capricious nor arbitrary cultural variation, but based in biological logic and mediated by socioecology. For example, the use of mother to refer to one's mother as well as aunts is common in many hunter-gatherers (Schefler 1978). People who use this terminology are not biologically naïve; they know the difference between their birth mother and their social mothers. These terminologies recognize the close genetic relationship children have to both their mothers and aunts, and reify equivalent support relationships that can be counted on, in high mortality or marginal environments (likewise, where paternity is relatively certain, father may be used to refer to one's father and uncles).

**Box 2.** Kin recognition.

Some abilities to recognize kin are quite ancient and shared across primates, while others arose more recently in the hominin line. At its most basic, giving birth at different times and in different locations is requisite to recognize one's mother, and hence one's siblings. Breeding and dispersal patterns have further effects on being able to recognize kin. Multi-generational maternal kin are identifiable if females are philopatric. However, recognition of paternal kin depends on some form of pairbonding or means to identify fathers and grandfathers. Human life history further extended the ability to recognize kin. Because the time to maturity is long and birth intervals are short, mothers often raise an infant, weanlings and juveniles simultaneously, which enables strong bonds to form between mothers and offspring, and between siblings (Chapais 2008). Complex kin systems, which are highly developed in traditional human societies, greatly expand the ability to distinguish a range of maternal and paternal relationships (social, step, biological and classificatory kin). Sophisticated means to recognize kin today includes genetic testing.

Kin selection is often invoked as the theoretical principle explaining cooperation and sharing within families. Hamilton (1964) elegantly simple formula offered an explanation for the widely documented empirical observation across species that individuals favor assisting kin, and close kin over distant kin. In its application to humans, kin-bias is particularly evident for childcare, which tends to be a family affair (Crittenden and Marlowe 2008; Henry et al. 2005; Kramer 2009; Leoneiti and Nath 2005; Scelza 2009; Weisner and Gallimore 1977). Likewise, a number of studies show that food is preferentially shared with relatives (Betzig and Turke 1986; Gurven et al. 2001; Koster 2011; Wood and Marlowe 2013; Ziker and Schnegg 2005).

While kin selection helps to understand why family members cooperate with each other, it does not explain why families emerge in the human line but not in other closely related species. Insight into why related individuals dependent on each other in ways not seen in other species can be drawn from the human life history of mothers raising multiple dependent young (see above), and the complexity of the human subsistence niche.

The complexity of human livelihoods means that only under rare circumstances does a person alone do everything needed to survive—a constraint that creates both opportunities and benefits for the division of labor (Kramer 2018). Simply said, there are insufficient hours in the day for any one individual to find food, procure and process it, make tools, construct clothing and shelter, care for children, and maintain social and information networks. Although many exceptions exist, a division of labor is an efficient means to solve the time allocation problem of not being able to do everything to survive, even if there are modest inequities, e.g., some individuals put more into the pot than they get out. Within families, divisions of labor occur across age, sex and skill. For example, from a young age, children are shared to with the expectation that they give back. Children may perform easier and less skilled tasks, but they produce other resources that both contribute directly to their own calorie requirements and are shared to others (Kramer 2005a, 2005b, 2011, 2014). Common children's activities include foraging for fruit and berries, digging small tubers, hunting for small game, fishing, collecting shellfish, harvesting grain, fetching water and collecting firewood, and also are the primary care-takers of their younger siblings (Kramer

and Veile 2018; Kramer 2021). Within families, a sexual division of labor complements an age division of labor (discussed in Section 3.2).

Kin selection also predicts conflicts of interest within families. While close ties within families theoretically link economic interests among its members, families are also composed of players whose agendas are not necessarily or always aligned. Within the family, conflicts of interest arise between males and females, parents and children and between siblings (Sulloway 2008; Boone 1988; Hagen et al. 2001; Lawson et al. 2012; Lawson and Mace 2008; Penn and Smith 2007; Strassmann and Gillespie 2002). Theory would predict that, although interests may be more aligned among those closely related than outsiders, within kin groups, an individual will ty to optimize his or her own survival, well-being or reproduction, even if it may be detrimental to other kin.

## 2.3. Reciprocity and Mutualism Foregrounding Multifamily Cohesion

For much of human history, we lived in small residential clusters of interacting families (Chapais 2008; Chapais 2011; Kelly 2013; Marlowe 2005a, 2005b). What tips humans from interacting with those outside the family as potential competitors to collaborators? Here, theories of reciprocity (Trivers 1971; Gintis 2000; Nowak and Sigmund 1998) and mutualism (Clutton-Brock 2002) developed in behavioral ecology help explain resource and labor pooling among nonkin and across families (Chapais 2001, 2006).

Although childcare, food sharing and coordinated labor activities are staples within families, ethnographers also often note that food commonly comes from outside the family (Gurven 2004a, 2004b; Gurven and Hill 2009; Hamilton et al. 2007; Hawkes et al. 2001; Hill et al. 2011; Kaplan and Hill 1985; Kramer and Ellison 2010; Murdock 1967; Wiessner 1982, 2002; Allen-Arave et al. 2008). As examples, among the Ache (foragers living in the forests of Paraguay), the majority of the meat one consumes is given from a nonfamily members (Gurven 2004a, 2004b). When a Savanna Pumé forager returns to camp with food, a portion is distributed to as many hearths as is practical (e.g., how many is in part determined by resource size; a caiman, for example can be shared out to many more families than a small basket of roots). The most common explanation for resource transfers among nonkin is that it mitigates risk and smooths day-to-day and individual variance in food supply (Gurven and Hill 2009; Kaplan et al. 2000; Gurven 2004a, 2004b).

In addition to food production, pairing skills across families figures in collaborations among nonfamily members (Chapais 2006). Partiality toward kin, yet preferences for competence plays out among the Savanna Pumé foragers. Hunting parties are often composed of relatives, but good hunters (e.g., those with high return rates) also seek out other proficient hunters as friends and hunting partners, a preference noted among other groups of hunter-gatherers (Wiessner 2002). Friendships also tie reciprocal relationships across unrelated families. Among the Yucatec Maya, for example, while households that help each other are usually are closely related, a portion of supportive helping relationships also are between households with no kin or affinal affiliation (Hackman and Kramer 2021, this volume).

Other undertakings that are part of normal subsistence or produce collective goods—building a house, constructing an irrigation ditch, hunting large or migratory prey, tanning hides, excavating a well—require coordination, if not cooperation across the group. These types of large-scale projects simply cannot be performed alone (Alvard and Nolin 2002 for the example of whale hunting). With increased technological complexity, examples become abundant.

Cheating, self-interest and other collective action problems destabilize reciprocity from taking hold in many animals. The capacity for reciprocity not to break down, and for humans to live in interdependent multifamily groups, also derives in part from the complexity of human subsistence, long human lives and intergenerational reputations. Many reasons have been forwarded why reciprocity might exist (Gurven 2004a, 2004b). However, collective action problems may themselves be muted because many different kinds of resources and labor are exchanged. How does one equivocate the five fish received

from one household, for the ten tubers they gave back? Or the value of the firewood received, for the fruit given in exchange? How do people value the time spent foraging for food for the time someone else spent processing it? Or that someone who is called upon to heal the sick, receives more meat than he contributed. In the modern setting, similar types of exchanges occur all the time. The point made here, is that the exchanges across many different currencies may stabilize reciprocity more than collective action models might predict. Generosity is highly valued among many hunter-gatherers (Marlowe 2010); if exchanges are grossly unequal sanctions are leveraged against cheaters. Because humans have life-long relationships and language to tell the story, reputations are inherited (Boehm 2012; Wiessner 2020), and cheaters do eventually pay.

## 3. Key Debates about Family Formation

What distinguishes the human family rests on species-specific patterns of mating and reproducing. These two domains, however, have generated substantial debate about the universality of certain traits and the ancestral conditions which gave rise to the modern family. Several of these debates are discussed below, with a focus on those that challenge conventional assumptions and have generated discussion on alternative approaches on the evolution of the human family.

### 3.1. Are Humans Patrifocal by Nature?

Debates about the origins of the family have surrounded which sex disperses at maturity, and whether families and multifamily groups are centered around maternal or paternal kin. The conventional male-centric view of both dispersal and residence has given way to more nuanced perspectives on family formation.

In many primate species, at sexual maturity, one sex stays, and the other emigrates and in time joins a new group—presumably in avoidance of incestuous breeding. Early human sociality is often described as male philopatric (meaning females dispersed at maturity) based on what chimpanzee do and what was presumed to be a universal hunter-gatherer dispersal pattern. This characterization persists in the nonanthropological literature, although it has long been disputed (Ember 1975; Lee 1979; Lee and DeVore 1968; Meggitt 1965; Murdock 1949; Turnbull 1965). In small human populations, dispersal varies considerably both individually and societally, with males migrating in some cases, females in others, both leaving home or neither leaving home in still others, and many groups expressing multiple patterns simultaneously (Alvarez 2000; Beckerman and Valentine 2002; Blurton Jones 2016; Kramer and Greaves 2011; Marlowe 2003b; Walker et al. 2011; Kramer et al. 2017). In other words, human dispersal is highly flexible.

Expectations about male philopatry led to further assumptions about patrilocality (a post-marital residence pattern, where women move to live with their husband's family) and patrilineality (descent and inheritance reckoned through the father and the male line) being the basis of human society (Radcliffe-Brown 1930; Service 1962; Steward 1955). Ethnographic reality, however, deviates from this expectation. A number of recent case studies point out that residence patterns among hunter-gatherers are flexible, facultative, and may change frequently across the life course (Marlowe 2010). Spouses often move between local groups, shifting residence between maternal and paternal kin throughout a marriage union (Alvarez 2000; Kramer et al. 2017; Marlowe 2004). For example, among the Hadza (Sub-Saharan hunter-gatherers), women prefer to live with their kin in the first years of marriage when they have young children to care for. As families mature, men prefer to live with their kin (Marlowe 2010, pp. 40–41; Wood and Marlowe 2011). Records of camp membership document that family composition varies with children's age and, moreover, may reassort daily, weekly and seasonally to include either maternal and paternal kin, a residence pattern referred to as bilocality or multilocality. Cross-cultural studies that synthesize much of the comparative hunter-gatherer data likewise show a preponderance of bilocal residence (Kelly 2013; Marlowe 2004; Kramer and Greaves 2011), disputing earlier expectations that the ancestral family is characterized by patrilocality and patrilineality.

Unless otherwise constrained, bilocality makes sense as the preferred residence pattern since it recognizes affiliations on both the mother's and father's side, maximizing the potential safety net options and alliances. Bilaterality is common in hunting and gathering economies where subsistence activities change throughout the year and bilateral networks permit greater geographic flexibility in residential mobility. In contrast, unilateral residential and descent kinship is more common among pastoralists and agriculturalists, who tradeoff excluding half their kin with building strong alliances with the other half (Van den Berghe 1990; Walker and Bailey 2014). At its simplest, at some demographic scale, coordinating with, sharing food, exchanging labor obligations and responsibilities with an ever-expanding group of people becomes impractical. While unilineality limits commensal group size and obligations, at the same time it creates large lineage affiliations for territorial defense, warfare (Chagnon 1979), water management and other large-scale collaborative projects.

### 3.2. Is Male Parental Investment the Driving Force in the Origin of the Family?

In few mammalian species do females and males parent together. The limitations usually cited are paternity confidence, e.g., males need to know that they are the father to invest in offspring, and that their help needs to make a difference to offspring quality. Consequently, habitual male care is typically associated with monogamy (social monogamy, if not biological monogamy), a combination that occurs in ~5% of mammalian species (Clutton-Brock 1991; Lukas and Clutton-Brock 2012). Although more common in the primate order (Lukas and Clutton-Brock 2013; Opie et al. 2013), male parental care is not observed in chimpanzees and bonobos, species to whom humans are most closely related.

Arguments about the evolution of paternal care in the hominine line are closely aligned with assumptions about the sexual division of labor. Here, the causal arrow is debated. The need for male help was traditionally argued to be the predominant pressure driving monogamy (Clutton-Brock and Harvey 1977; Kleiman 1977), while recent research makes a strong case that paternal investment is a consequence of monogamy (Boomsma 2009; Lukas and Clutton-Brock 2013; Opie et al. 2013). No doubt a division of labor is an efficient means to manage a household and raise young (Becker 1981; but see Sear 2021). Although fathers generally help little with childcare, they are important economic contributors in many contemporary contexts (Gurven and Hill 2009; Kaplan et al. 2000; Marlowe 2007), both in traditional and industrialized societies, and their help is linked to improved survivorship and well-being (Gurven and Hill 2009; Hill and Hurtado 2009; Kaplan and Lancaster 2000; Lancaster et al. 2000; Marlowe 2003a, 2003b; Meehan et al. 2013; Quinlan and Quinlan 2008). However, as an evolutionary argument, the focus on male investment may oversimplify reasons why families formed in the first place. The division of labor literature overwhelmingly has centered on male specialization and hunting. However, this narrowly focuses on a food that is too ecologically variable as a dietary constituent to be broadly explanatory, and ignores the many other ways that the division of labor figures into daily lives. For example, for Savanna Pumé foragers, terrestrial game constitutes approximately 5% of the diet, and rather than being about meat, the division of labor is driven by women's foods (fruit and roots), food processing and domestic tasks. Focus on the sexual division of labor as the impetus for family formation also overshadows the importance of the age division of labor, which is critical in incorporating older and younger generations into family groups (Kramer 2011).

### 3.3. Cooperative Breeding as an Alternative Evolutionary Basis for Family Formation

Mothers, or mothers and fathers, are often challenged to alone raise the multiple dependents that are characteristic of our life history. For example, where resource flows have been observed in Maya families, fathers contribute as much time as mothers to economic activities that support children (Kramer 2009). Nonetheless, when parents have more than four children, which they often do under natural fertility conditions, the work effort of mothers and fathers is insufficient to meet family consumption. In the Maya case,

their children's help filsl this gap (Kramer 2005a, 2005b; Lee and Kramer 2002). Recognition that others help parents (often their own children, older generations or collateral kin) led to an important recent shift in thinking about family systems. Rather than a focus on the pairbonded parent–child unit, cooperative breeding recognizes that many caregivers help mothers in addition to fathers (Hrdy 1999; Ivey 2000; Kramer 2005a, 2005b; Kramer 2010; Mace and Sear 2005).

Cooperative breeding is an unusual reproductive system in which group members other than parents help to raise offspring who are not their own. Although expressed in diverse taxa, cooperative breeding is rare, occurring in an estimated 9% of bird (Cockburn 2006) and 3% of mammalian species (Russell 2004). Allocare situationally occurs across a range of primates, however, cooperative breeding is not a shared great ape trait (Hrdy 2005; Hrdy 2016; Lancaster and Lancaster 1983). Hence, its emergence in the human lineage marks a significant departure in parenting strategies and may signify an alternative pathway to family formation.

Cooperative breeding unites many traits that characterize human modernity: life history, pooled energy budgets, family and group structure (Kramer 2010; Kramer and Ellison 2010). Distinct theoretical arguments have been made for the evolutionary importance of juvenile helpers (Kramer 2011, 2014; Kramer and Otárola-Castillo 2015) and grandmothers (Alvarez 2000; Hawkes 2003; Hawkes et al. 1989; Hawkes et al. 1998; O'Connell et al. 1999). The help mothers receive benefits them by alleviating time constraints, which arise from supporting multiple dependents of different ages—infants, young children and old children, who require different kinds of resources and time investments. For example, Maya mothers who raise large families only provide approximately 40% of her family's consumption, and approximately 50% of infant care, the balance of which is met by the help of her children and husband (Kramer 2005a, 2005b; Kramer and Veile 2018). In managing the competing demands of multiple dependents, mothers in many societies find the extra time to devote to young children, particularly nursing children, by downwardly adjusting their investment in economic activities—foraging for food, time spent in agricultural work, domestic activities or wage employment, depending on their livelihood (Hawkes et al. 1997; Hames 1988; Hurtado et al. 1985; Hurtado et al. 1992; Kramer 2009; Marlowe 2005a, 2005b). The help mothers receive also has a demonstrated positive effect on their children's health, growth, and well-being (Kramer 2010, Table 2), as well as increasing maternal fitness primarily through enabling mothers to give birth at shorter intervals and improving child survivorship (Kramer 2009; Lahdenpera et al. 2004; Lee and Kramer 2002; Sear and Mace 2008).

In addition to her own children and mother, aunts (usually mother's sister), grandfathers, other relatives and nonrelatives may also help to provide childcare, food, shelter and other assistance. Human children are well adapted to having nonparental helpers, so much so that they have positive effects on children's development. Exposure to multiple caretakers expands a child's social sphere and is associated with cognitive and psychological benefits (Pope et al. 1993; McKenna 1987; Wilson 1986; Weisner and Gallimore 1977; Isler and Schaik 2012), such as emotional regulation and shared intentionality, which is the basis for theory of mind and other uniquely human prosocial abilities (Burkart et al. 2009).

## 4. Conclusions

In sum, the evolutionary arc of family formation illuminates how unusual the human family is as a social form and the diversity its shape takes. Because individuals commonly maintain life-long relationships with their natal families, despite marriage, emigration and establishing families of their own, it sets the stage for expanding spheres of social interaction. While the nuclear family norm of parent(s) and dependent children prevails in industrialized societies today, it was likely rare in the past. A behavioral ecology approach to family studies is a useful framework to appreciate family diversity past and present and reconsider patrifocal views of how families form.

Human life history and the central dilemma of mothers—how to find enough hours in the day to support dependent offspring—is foundational to understand why cooperative relationships between mothers and children, spouses and others emerged in the human line, and no doubt resonates with mothers today. In post-industrial societies, mothers face new challenges to the same allocation problem of providing competent childcare while finding time for economic and domestic activities—a tradeoff as relevant today as it was in the past. As a theoretic starting point, kin selection, reciprocity and mutualism allow us to square preferences within families, and why family members are often more willing to cooperate with each other than outsiders. It also allows us to understand why there may be conflicts, and why the family is often not enough and we live in communities of multifamily groups. Human-specific features of family life such as the age and sexual division of labor, pairbonds, sharing, reciprocity and cooperative breeding can be comprehended in the context of the complexity of human livelihoods in both traditional and industrial societies. The cross-cultural empirical record supports that the family is a highly flexible social organization that is transiently, culturally and ecologically adaptable, adynamic less transparent from traditional positions on patrilocality, patrilineality and male parental care. This is not to say that these features of family formation are not evident and important in the human record, but other perspectives such as cooperative breeding may further a more inclusive perspective.

**Funding:** This research received no external funding.

**Conflicts of Interest:** The author declares no conflict of interest.

## Notes

1   For simplicity *industrialized* is used throughout to refer to contemporary industrialized nation-state societies, in contrast to *small-scale* or *traditional societies*. Small-scale society here refers to small rural communities, often indigenous, or culturally, ethnically homogeneous, that while existing today within nation states have autonomy in terms of economy and governance and rely largely on local production for subsistence. Defining small-scale societies has been approached from several perspectives: (1) demographically, they are small communities; (2) they have minimal connection to market forms of energy and national supply chains; (3) their social and informational networks are predominantly local and noninstitutional. *Nonindustrial* is sometimes used as the preferred term, however it suggests a historical trajectory and takes the perspective of "us compared with them," rather than the other way around, which is more evolutionarily appropriate. Unfortunately, no term describes either society fully, in its variation or without bias.

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
