# Peer review of "The Human Family—Its Evolutionary Context and Diversity"

_socsci, doi:10.3390/socsci10060191_

Round 1

Reviewer 1 Report

General comments:  wonderful review of human behavioral ecology.

I suggest using a different term than "biological family" in the introduction [line 21]. All families are 'biological', and the apparent specific meaning here is not provided. 

The concept of societies [1.1] composed of "nested families" and "nested coalitions" is discussed at some length in Flinn et al 2007.

Not sure if I agree with their conclusions, but a relevant discussion of the Chimp vs. Gorilla model for the family [1.4] is presented in Geary & Flinn 2001. 

I view a key aspect of "behavioral ecology" [2] as its basis in phenotypic plasticity (West-Eberhard 2003).  Behavior varies in adaptive response to ecological conditions.

[line 243] extra space after cites.

[line 249] perform

[line 341] delete "a" family members...

The potential importance of coalitions and alliances for cooperation and reciprocity could be discussed in [2.3]; e.g., S. Bowles, R.D. Alexander, N.A. Chagnon

[2.4] Early research papers on cooperative breeding in a human society include Flinn, 1989; Turke, 1988.

[line 498] traditional

[line 502] this is a good place to reiterate the strength of behavioral ecology to predict cultural and individual variations, such as what conditions favor different mating and family relationships...

[line 603] 'post-industrial' includes a wide range of family relations, including multiple-generation female kin.

Flinn, M.V. (1989).  Household composition and female reproductive strategies.  In: Sexual and reproductive strategies, A. Rasa, C. Vogel & E. Voland (Eds.), pp. 206-233.  London: Chapman and Hall.

Flinn, M.V., Quinlan, R.J., Ward, C.V., & Coe, M.K. (2007).  Evolution of the human family: Cooperative males, long social childhoods, smart mothers, and extended kin networks. In:  Family relationships, C. Salmon & T. Shackelford (Eds.) Chapter 2, pp. 16-38. Oxford: Oxford University Press.

Geary, D.C. & Flinn, M.V. (2001).  Evolution of human parental behavior and the human family.  Parenting: Science and Practice, 1 (1&2), 5-61.

Turke, P. 1988. Helpers at the nest: childcare networks on Ifaluk. In: Betzig, L., Borgerhoff Mulder, M., & Turke, P., editors. Human reproductive behavior. Cambridge: Cambridge University Press. pp. 173–188.

Author Response

Reply to Reviewer 1

Dear Reviewer, Thank you for your comments.  I have made all of the suggested changes, added the requested references, and corrected the typographic errors.  Because two sections were deleted (in response to another reviewer), a few of the comments no longer pertain. Of note, as another reviewer commented, I replaced ‘biological family’ with ‘conjugal’ or ‘nuclear ‘family throughout. Although I used these terms interchangeably, I can see why that would be confusing.

Reviewer 2 Report

Although the authors state the purpose of the paper according to the lines 59-60: "the emphasis is here on small-scale societies" but nevertheless the title of the paper remains a general one that can even generate confusion on the topic, and it is necessary to adapt the title to the content of the paper.

The abstract doesn't fit into the publication requirements of the journal, being a summary, which requires completions in accordance with them, respectively to refer to the context of the paper, methods used, summary of main results and possibly a brief conclusion on the paper. See the "Instructions for Authors" section of the journal presentation.

The paper does not contain all the sections in accordance with the requirements of the journal, respectively it does not have the section on the methods used and the research questions proposed by the authors are not defined separately, and if it is possibly to indicate what are the limits of the research presented by the authors.

Further, I reproduce the structure given by the journal requirements, in order to arrange the material according to these requirements, respectively "Research manuscript sections: Introduction, Results, Discussion, Materials and Methods, Conclusions (optional)".

Excessive references are used, with references to works recorded in the list of references, and the possibility of verifying these citations is reduced, by the sometimes general approach to the cited references, used by the authors in the paper. I recommend a new analysis of the references used and can be maintain which are in the context of the paper.

I recommend the major revision of the paper in order to improve the presentation of the material and the compliance with the mentioned publication requirements.

Author Response

Dear Reviewer, Thank you for your comments.  I have now stated clearly in the introduction that this is a review paper, which I hope obviates the concern that the paper doesn’t contain a Materials & Methods and Results section.  I changed the title of the paper.  I have also altered the abstract, and the conclusion. While I have reduced the number of sections and therefore references, as a literature review paper, there are still a fair number of references to give credit to the history of certain ideas.    I am happy to further reduce citations if necessary. 

Reviewer 3 Report

This paper presents a review of the unique traits of human families, compared to our non-human relatives and highlights the contributions of HBE to the study of the family. The review is well researched and thorough.

My primary concern is that in its current form it is hard to identify a central thesis or argument. Many, many concepts are covered, but often too fleetingly for a reader from a non-HBE background to follow. I think the paper would benefit from greater streamlining centered around a main argument, even if some of the current breadth is sacrificed.

My other main comments center on two main observations: the paper is written assuming that the reader (1) comes from the same socioecological background as the author(s) and (2) that the reader is well-versed in HBE. I think the paper would be improved by addressing the positionality of the researcher(s) and adding clarity to the framing for non-HBE researchers.

Regarding the first point, throughout the author(s) refers to ‘our own societies’ (e.g. lines 56, 63) or ‘our experiences’ (line 135). Please explicitly state what you are referring to rather than saying ‘our’ because 1) I don’t know what society the author(s) lives in and 2) your readers may or may not live in the same society as you. I’m assuming you are talking about North American and/or European market integrated contexts. Even within these contexts though, I’m guessing the reference is to higher-SES family norms not norms in low-SES groups within high-income countries (e.g. nuclear families as opposed to multi-generational households). Just as you have defined ‘traditional societies’ I would define what you mean by ‘our own’ society rather than assuming a shared experience with your readers.

With regards to the second point, I think greater care needs to be taken to consider what terms and concepts are widely used in the social sciences and those specific to HBE. Throughout there are vocabulary and ideas that may be unfamiliar to non-HBE social scientists that are worth introducing/defining more carefully to make the paper more accessible to a general readership. (e.g. philopatric, fast/slow life history, inclusive fitness). Some more specific points:

  • I’d like an expanded introduction to why you are comparing ‘traditional’ societies to non-humans. Within HBE this is the norm, but to a non-HBE person they may think that you are assuming that ‘traditional’ societies are relics of the past – and human/chimpanzee comparisons are sometimes perceived as offensive. Traditional societies are of course not relics of the past, but it is worth more explicitly addressing this misconception head-on in explaining why comparing these pops to human ancestors & non-human primates remains common in and fruitful to HBE research.
  • There are several places where general statements need to be backed up with demonstrations, rather than just lists of examples. Two examples:
    • On lines 289-295 you introduce conflict between kin members and list some of the key players in these conflicts. It’d be helpful to lay out the logic of one or two of these types of conflict – or even just the general logic of conflict between family members (i.e. interests are more aligned between kin members than with outsiders, but within kin groups each person will try to optimize their own fitness even at detriment to other kin members). To someone less well versed in HBE this paragraph may be lacking key details.
    • Similarly, in discussing cooperative breeding at one point you list two implications, but it would be helpful to give examples to demonstrate these implications (lines 440-443) rather than just saying they exist.

Other comments:

  • I’d like a slightly more descriptive abstract that outlines the primary arguments/take home messages of the paper.
  • Line 21: it may be clearer to say nuclear family – I think that is what you mean by biological?
  • Line 23: this doesn’t seem like a ‘yet’ statement. Maybe just delete sentence as your next paragraph covers it.
  • The conclusion feels a little bit out of the blue – a lot of info about postindustrial societies is introduced for the first time as is the concept of the demographic transition. I actually think this paragraph (with a bit of an expansion) would be better placed in the introduction.

Author Response

Dear Reviewer, thank you for very helpful comments. With some distance, I agree that the review was too broad, and have made a number of changes to center the focus.

 1. In terms of a central thesis, my goal in this review was to cover different points of view, not make an argument per se, or push forward my particular position. However, I certainly see the value in having a tighter focus, and found this comment most helpful. In the revision, I state the goals of the paper in the abstract and introduction.   

 2. I streamlined around these three stated goals by i) deleting the section 2.4 Parental investment & cooperative breeding. ii) Restructuring Section 2 around behavioral ecology approaches to within family dynamics (kin selection) and between family dynamics (reciprocity and mutualism); this also involved some organizational changes. iii) In Section 3, I deleted the first subsection Are Humans monogamous or polygynous, as being one of the sections that could be forfeited to “ streamline around a main argument even if some of the current breadth is sacrificed”. iv) throughout the manuscript I deleted a number of side thoughts.

Other changes in response to the suggested comments...

  1. Agreed that verbiage such as ‘our own society’ doesn’t work in this context. I wasn’t referring to any norms or making assumptions about any specific kind of ‘our society’, except that most readers are likely live in nation states with centralized governments, low fertility and low mortality—it was that contrasted I wished to make. It was intended to bring in readers that may not be accustomed to HBE approaches or who themselves don’t work in small scale societies by making personable. I see how it could be construed otherwise and I took out all reference to ‘our’ or most references to ‘we’. 

  1. Generally I tried to eliminate as much use of special language as possible. I added a statement in the introduction, that specialized language was italicized and briefly defined, and defined the words you suggested as well as a few others.

  1. I coalesced passages about why I was using examples from small-scale societies by i) adding a paragraph to the end of the introduction, and including in that much of the conclusion (also as per your suggestion), which described what is demographically salient about postindustrial populations to family formation. This does work much better, thank you for the suggestions. ii) there is also a footnote defining small-scale and industrialized societies, with the caveat that these are imperfect terms. iii) Yikes, I definitely don’t think of small-scale societies as relics of the past, and have actually railed about that perspective, so thank you for pointing that for this audience, I might need to say this explicitly, so have added such a statement to the introduction.

  1. I edited the abstract to be more explicit.

7.I took out the use of ‘biological family’

  1. I made all the other suggested changes to wording.
  2. The reviewer requested that general statements need to be backed up with demonstrations, rather than just lists of examples; these changes were made.

Reviewer 4 Report

Overall, this review of the evolution of the human family is thorough and will serve as a wonderful source for behavioral ecologists and non-behavioral ecologists alike. While I have a number of suggestions, they are all relatively minor in nature.

Minor revisions:

Lines 4 and 21: The term “biological” family without definition would presumably include all families based on consanguineal relationships. Therefore, “nuclear” family would be clearer to use here until “biological” family is defined.

Lines 56 and 63: I would discourage the use of “our own societies” as it presumes a readership. While the most agreed-upon terms seems to be in-flux, I would suggest “developed,” “wealthier,” or “nation-state.”

Line 107: Define philopatric here, as this is not an anthropology journal.

Line 112: While it’s hit upon below, I think it would be good to explain why multi-group relationships are more important in humans. As highlighted in the Chapais works, this is primarily that a) unlike other primates, humans maintain relationships with kin from previous groups after dispersal, and b) humans are aware of and maintain relationships with both maternal and paternal kin.

Line 129: Again, you might want to define “consanguineal” and “affinal”.

Line 250: I would add the hedge, “For some mobile hunter-gatherers…”

Line 297: This paragraph seemed out of place, but I believe it’s meant to segue to the next section. I believe it would be easier to follow if it simply was placed under the 2.3 heading.

Line 298: The distinction between kin-selected “altruism” and mere ostensible altruism, which actually pays out in the long run due to the pathways suggested by reciprocal altruism or indirect reciprocity, is likely to be lost on a non HBE readership. This might need to be more thoroughly fleshed out there.

Same paragraph: Given the net downward resource flows toward descendants, and the clear evolutionary logic for parental investment, I find it hard to conceptualize parental investment to non-infant children as primarily (or even substantially) contributions to a reciprocal relationship as it seems to be presented here. Similarly, in line 307, the phrase “From a young age, children are shared to with the expectation that they give back,” rings untrue. There is an expectation that children contribute, but without significant data to back it up, I find it hard to believe that parents give food to their five-year olds primarily because of the returns they expect from the reciprocal relationships, and not because of an evolved sense of parental concern. If children do not contribute back, would parents really stop investing? My home life suggests not!

Line 333: You might want to add Nowak & Sigmund, 1998 and Gintis, H. 2000 to cover indirect reciprocity and strong reciprocity (although strong reciprocity is more aligned with kin selection).

Line 485: Testes size and sexual dimorphism should be contextualized in relation to other great apes.

Line 495: “Monandry” seems exclusive to “polyandry”.

Section 3.1: In the last paragraph the authors do a pretty good job of highlighting what can be stated. However, a bit more could be stated following the behavioral, cultural, and anatomical evidence. Namely, that humans are oriented less toward high levels of male-male competition associated with high payoffs to winning contests to monopolize women compared to other polygynous great ape species (moderate SD, female autonomy in partner selection to varying but often substantial levels), that there are lower levels of promiscuity compared to multi-male, multi-female chimps/bonobos (long-term relationships with multiple reproductive events, cross-cultural understandings of jealousy, sexual preferences/restrictions in pairbonds, moderate teste size, lack of copulatory plugs, etc.), and that there is biparental care of shared children. This all suggests a shift in the direction of a monogamous, bi-parental mating system, but not entirely so. These systems need not be conceptualized categorically.

Grammatical

Line 57: There is an errant “1”.

Line 96: The second and third commas seem superfluous.

Line 117: “Kaplan, 2000 #91” appears twice.

Line 155: It seems the hyphen there should be an em dash.

Line 212: The last comma should probably be an em dash.

Line 216: Sentence frag.

Line 561: You are missing the closing parentheses.

Line 562: The sentence has a “both…” but then lists only one thing.

Line 575: “driven BY womens’s…”

Line 583: Similarly, the “three distinct evolutionary steps” are ambiguous. It seems like this is the integration of 1) juveniles, 2) grandmothers, and 3) fathers, but this could be made clearer.

Nowak, M. A., & Sigmund, K. (1998). Evolution of indirect reciprocity by image scoring. Nature, 393(6685), 573-577.  Retrieved from <Go to ISI>://000074150100051

Gintis, H. (2000). Strong reciprocity and human sociality. Journal of Theoretical Biology, 206, 169-179.

Author Response

Dear Reviewer, Thank you for your comments and suggestions, all of which I made. I appreciate your time and effort, and believe the revised version benefited from the suggestions.

  1. I replaced the term ‘biological family’ with ‘conjugal’ or ‘nuclear’ family. I use the terms interchangeably, but I see where this is confusing.

  1. As per other reviewers, I eliminated use of ‘our’ and ‘we’. My intention was only to create a contrast between nuclear and nonnuclear social organization at the societal level.   But agree, that  it doesn’t work in this context.

  1. I defined ‘philopatric’, ‘consanguineal’, ‘affinal’ and a few other specialized words.

  1. I made all the typological corrections -- thank you for catching.

  1. comment for ln 112 -- I made an adjustment to the language, so this should be clearer up front.

  1. As per another reviewer’s comments, I deleted two sections to hone the article’s focus, so that the suggested changes in ln 297-298, ln 485 and ln 495 no longer apply. But I agree with your comment (e.g. why the sections were eliminated).  This also pertains to the comment about section 3.1, which I deleted – agreed much more could have been made of this, but the article is already long, I’ve previously published at length on this topic and it seemed like the most sensible section to delete to better focus the manuscript.

  1. I added the suggested references.

Round 2

Reviewer 2 Report

If the authors write in the Abstract and "in the introduction that this is a review article", then the type of paper must be changed from Article to Review, according to the types of publications announced by the journal.

Author Response

Dear Reviewer, I believe one of the editors changed the article type (I don't think that I can do from my end),  as type now appears as 'review'.   thank you for your comments. 

Reviewer 3 Report

Thanks for the thoughtful responses to my previous comments. I appreciate the streamlining and additions to the set up for the paper which more clearly articulate the paper's goals and structure. I also like the new title.

I have some lingering concerns about how dense certain sections are. These dense areas could use some unpacking. For example, the last paragraph of section 3.3 on cooperative breeding introduces a lot of new information (i.e. that exposure to multiple caretakers is associated with cognitive and psychological benefits and have implications for development of prosociality) accompanied by many citations. It would be really helpful to have some expansion on what that all actually means - could use an example or two from those citations to show the reader what you mean? 

The paragraph above that one (second to last in section 3.3) is another good example of a paragraph that could use unpacking. 

Author Response

Dear Reviewer, Thank you for your additional comments.  I have made the suggested changes by editing and adding material to the last two paragraphs of section 3.3, which hopefully sufficiently unpacks the ideas.  In the last round I tried to significantly shorten (and did by ~1200 words) the manuscript, so didn’t want to add back in too much length. But I think the changes make cooperative breeding more accessible to a general social science readership.